# Bibliometric Analysis of Research Trends and Global Collaborations in Anesthesia on Neuromuscular Blockers and Antagonists (2000–2024)

**DOI:** 10.3390/healthcare13101146

**Published:** 2025-05-14

**Authors:** Turan Evran, Hüseyin Özçınar, İsmet Çopur, Beliz Bilgili

**Affiliations:** 1Department of Anesthesiology and Reanimation, Faculty of Medicine, Pamukkale University, 20070 Denizli, Turkey; icopur@pau.edu.tr; 2Department of Computer Education and Instructional Technology, Faculty of Education, Pamukkale University, 20070 Denizli, Turkey; hozcinar@pau.edu.tr; 3Department of Anesthesiology and Reanimation, School of Medicine, Marmara University, 34899 İstanbul, Turkey; belizbilgili@gmail.com

**Keywords:** anesthesia, bibliometrics, cholinesterase inhibitors, neuromuscular blocking agents, neuromuscular depolarizing agents, neuromuscular nondepolarizing agents, research, trends

## Abstract

(1) Background: The aim of this bibliometric study is to analyze global research trends, citation impact, and scientific collaborations in the field of neuromuscular blockers (NMBAs) and their antagonists between 2000 and 2024. (2) Methods: Data were retrieved from the Web of Science Core Collection (WoSCC) using Boolean search strategies. Bibliometric analyses were conducted using R bibliometrix, VOSviewer, and CiteSpace software to visualize collaboration networks, keyword trends, and citation bursts. (3) Results: A total of 499 articles were analyzed, with the United States of America (USA), China, and South Korea leading in productivity, while France had the highest citation impact. Influential authors included Mertes PM and Fuchs-Buder T. Emerging topics such as sugammadex, sevoflurane, and neuromuscular monitoring were identified, reflecting a shift from pharmacokinetic studies to safety and monitoring strategies. (4) Conclusions: The findings indicate a marked increase in studies on neuromuscular monitoring and reversal agents, such as sugammadex, over the past two decades. The USA, France, and China emerged as the most contributory countries in NMBAs research, with their extensive international collaborations playing a pivotal role in shaping scientific progress. Highly influential studies have predominantly focused on NMBA pharmacokinetics, safety, anaphylaxis risks, and the clinical benefits of sugammadex, underscoring its critical role in reducing residual neuromuscular blockade (rNMB) and enhancing patient safety.

## 1. Introduction

Neuromuscular blocking agents (NMBAs) have been utilized in clinical practice for approximately 80 years as a critical component of modern anesthesia [1]. With the initial clinical introduction of d-tubocurarine by Harold Griffith and Enid Johnson in 1942, the impact of muscle relaxation on both surgical success and patient safety became clearly understood, fundamentally reshaping anesthesia practice by integrating anesthesia, analgesia, and muscle relaxation [2]. Following this milestone, the development of gallamine in 1947 paved the way for the clinical use of numerous new agents, including succinylcholine, pancuronium, atracurium, and rocuronium [3,4]. Due to their advantages in stabilizing the surgical field, preventing involuntary muscle movements, and facilitating mechanical ventilation, NMBAs have become indispensable in contemporary anesthetic practice.

Depolarizing NMBAs are particularly favored for short-duration procedures because of their rapid onset and short action, whereas non-depolarizing NMBAs are widely used due to their longer duration and reversibility [5]. However, these agents may cause complications such as allergic reactions, cardiovascular side effects, and prolonged neuromuscular blockade, necessitating careful management in clinical settings [6,7]. The use of acetylcholinesterase inhibitors to reverse the effects of NMBAs and, notably, the introduction of sugammadex for steroidal agents have emerged as significant innovations in modern anesthesia [8]. Moreover, objective methods such as Train-of-Four (TOF) monitoring, which assesses the depth and recovery of neuromuscular blockade, play a critical role in preventing postoperative complications [9,10].

In recent years, research on the clinical applications of NMBAs has expanded into new areas. Multidisciplinary topics such as acute respiratory distress syndrome (ARDS), intensive-care-related muscle weakness, and inflammatory responses have demonstrated that NMBAs play a vital role not only in surgical anesthesia but also in intensive care settings [11,12].

Bibliometric analysis is a powerful method for quantitatively and qualitatively evaluating academic literature and has become increasingly common with the advent of digital databases [13,14]. Through techniques such as performance analysis and scientific mapping, it is possible to identify research collaborations, gaps in the literature, and future research directions [15,16]. Typically, these analyses are conducted using large scientific databases, like Web of Science and Scopus, which reveal thematic research areas and keyword co-occurrences [17,18]. Although numerous clinical studies on NMBAs and their antagonists have been published, there is a significant gap in comprehensive analyses that systematically assess global research trends, international collaborations, and developmental dynamics in this field. In addition to mapping academic contributions, such analyses can highlight clinically relevant trends, thus helping anesthesiologists to prioritize evidence-based practices and identify future clinical research areas. To our knowledge, there has been no bibliometric analysis research conducted in this field. This study aims to evaluate the research on neuromuscular blocking agents and their antagonists in the field of anesthesia from 2000 to 2024 using bibliometric analysis. Utilizing data from the Web of Science Core Collection, this analysis comprehensively examines global collaborations, research trends, and scientific productivity. The primary objective is to elucidate the current status and scientific dynamics of the field, identify prominent research topics, and provide guidance for future research directions.

## 2. Materials and Methods

### 2.1. Data Collection

In this study, scientific publications related to NMBAs and their antagonists in the field of anesthesia were retrieved using the Web of Science Core Collection (WoSCC) database. WoSCC was chosen as the sole data source for this bibliometric analysis, consistent with methodological approaches in similar studies [13]. This platform exclusively provides access to journals indexed in the Science Citation Index Expanded (SCIE), which is widely recognized for its high-quality and standardized indexing structure. The comprehensive and consistent nature of WoSCC’s bibliographic records ensures the reliability of the citation analytics and bibliometric data, making it the preferred database for systematic bibliometric research in medical and scientific disciplines.

All data used in the study were obtained from publicly available sources and, since they did not contain personal information, did not require ethical committee approval. The data collection process was carried out on 20 September 2024, using the advanced search options of WoSCC. The search was executed with the assistance of Boolean operators, combining keywords in the “Title” (TI) and “Author Keywords” (AK) fields with both “OR” and “AND” operators. The selected keywords were grouped into Keywords 1 and Keywords 2, with each group internally combined using the OR operator. Then, the two groups were merged using the AND operator to construct the final Boolean search query (Table 1). Only articles or reviews published in English between 2000 and 2024 and indexed in the Science Citation Index Expanded (SCI-Expanded) were included in the study. A list of manually selected articles is provided in the Appendix A.

To maintain consistency in keyword interpretation and reduce variability related to language differences in bibliographic records, only English-language articles were included. Keywords were selected through expert consensus among anesthesiologists with over 10 years of experience, ensuring they were both specific and comprehensive to reflect the scope and current trends in the field. Publications in veterinary sciences, duplicate articles, studies with incomplete bibliographic information, and those published in languages other than English were excluded. All the data obtained were saved in plain text format for the subsequent analysis.

### 2.2. Data Analysis and Visualization

The collected bibliometric data were analyzed and visualized using the R bibliometrix package (version 4.4.1), VOSviewer (version 1.6.20), CiteSpace (version 6.3.R1), and the Python 3.12.2 programming language with the sklearn and matplotlib libraries. Initially, the data retrieved from the WoSCC database were organized using Microsoft Excel, and column–graph combinations were generated for graphical representation.

The changes in publication counts over the years were evaluated using linear regression analysis with the Python sklearn and matplotlib libraries. The independent variable was the year (2000–2024), and the dependent variable was the annual number of publications. R^2^ and *p*-values were calculated to assess the statistical significance of the model. The analysis results were visualized using the matplotlib library, with data points depicted as blue circles and the overall trend shown with a red line.

### 2.3. Bibliometric Analyses

Quantitative analyses of publication output and citation impact for countries, journals, and authors were conducted using the R bibliometrix package. In these analyses, countries, journals, and authors were ranked according to their total publication counts (productivity) and citation metrics (scientific impact), with frequency distributions calculated for each category. The top 10 most productive countries, journals, and authors were presented in a table showing the total publication count (N) and percentage distribution.

Network analyses were performed using VOSviewer to visualize the collaboration and relational networks among countries, journals, and authors in neuromuscular blocker research. Specifically, the “Country Collaboration Network in NMBAs Research” was used to examine collaboration relationships among countries; the “Journal Co-Citation Network in NMBAs Research” to identify co-citation relationships among journals; and the “Co-Authorship Network in NMBAs Research” to analyze academic collaboration links among authors.

In these analyses, nodes represented countries, journals, or authors, with the node size proportional to the number of publications associated with each entity. Colors represented research groups obtained from clustering analysis, and the connecting lines indicated instances of co-citations or collaborative publications. The thickness of these links reflected the number of jointly authored or co-cited articles.

### 2.4. Thematic and Trend Analyses

To identify the key themes and trends in neuromuscular blocker research, document co-citation, reference burst detection, and keyword citation burst analysis were conducted using CiteSpace. These analyses included the following approaches:

Co-Citation Analysis: “Neuromuscular Blockers Based on Co-Citation Analysis” was performed to examine the thematic evolution within the research field.

Top References: “Top 20 References with Strongest Citation Bursts (2000–2024)” was used to determine the most influential publications over time.

Keyword Burst Analysis: “Citation Burst Analysis of Top 25 Keywords in NMBAs Research” was performed to analyze the 25 keywords that had gained popularity over time.

Using the R bibliometrix package, the most frequently used keywords in neuromuscular blocker research and their co-occurrence frequencies across various publications were analyzed. The relationships among frequently co-occurring keywords in the literature were mapped using the “Keyword Co-Occurrence Network in NMBAs Research” analysis.

Furthermore, a “Sankey Diagram of Keyword Evolution Across Time Periods” analysis was performed to visualize temporal shifts in keyword usage frequencies over defined intervals. In these analyses, node sizes were proportional to the frequency of keyword usage in the literature, with colors indicating the research theme to which the keyword belonged and link thickness representing the number of times two keywords appeared together in different studies.

### 2.5. Centrality and Clustering Metrics in Co-Word Analysis

In this study, co-word (keyword co-occurrence) analysis was applied to networks where nodes represented keywords. To evaluate the roles and influences of these keywords within the network, we used centrality metrics such as PageRank centrality, betweenness centrality, and closeness centrality, which are widely applied in bibliometric network analyses [19]. PageRank reflects the importance and frequency of connections a node receives; betweenness indicates the node’s role in connecting different clusters; and closeness represents how centrally a node is positioned within the network.

To identify thematic subdomains, we applied the Leiden clustering algorithm, which effectively detects communities in large and complex datasets by grouping keywords with similar topical patterns [20]. Nodes grouped in the same cluster represented thematically cohesive areas of research.

## 3. Results

### 3.1. Publication Outputs in Neuromuscular Blockers and Antagonists Research

In this study, a total of 727 articles published between 2000 and 2024 in WoSCC database were analyzed. During the screening process, 143 studies were excluded because they were not of the article or review type, 51 studies were excluded for not being published in English, and 34 studies were excluded because they belonged to the veterinary sciences category. Consequently, 499 articles that met the inclusion criteria were included for detailed analysis (Figure 1). These publications represent studies from 51 different countries, published in 163 different journals. Contributions came from 2424 different authors, and 185 studies received funding support.

Analysis of publication and citation trends in NMBAs research from 2000 to 2024 demonstrated a general upward trend in annual publication numbers (Figure 2). The citation data in Figure 2 represent the number of citations received in each specific year.

A linear regression analysis of the number of publications from 2000 to 2024 showed that the year variable explained 35.8% of the variance in publication count (R^2^ = 0.358, adjusted R^2^ = 0.329). The regression model was found to be statistically significant (F = 12.28, *p* = 0.002). The resulting regression equation (Y = 0.6283X − 1243.0384) indicates an average annual increase of approximately 0.63 publications. The coefficient was significant within the 95% confidence interval [0.256, 1.000] (Figure 3). Nonetheless, the growth trajectory was heterogeneous; marked increases occurred during 2010–2015 and 2017–2021, whereas declines were observed during 2005–2010 and 2022–2024.

### 3.2. Analysis of Countries

The scientific productivity and impact of countries in research on neuromuscular blockers and antagonists were evaluated based on publication and citation counts (Table 2). The United States of America (USA)—with 78 articles, accounting for 15.63% of the total—was identified as the most productive country, followed by the People’s Republic of China (10.02%) and South Korea (9.21%), which ranked second and third, respectively.

Regarding citation counts, France emerged as the most influential country, with 1776 citations, while the USA and Germany followed with 1055 and 908 citations, respectively. Notably, Turkey appears on both the list of most productive countries (7.81%) and the list of countries with the highest citation counts (3.30%). To delineate inter-country collaborative networks, a Country Collaboration Network analysis was performed using VOSviewer.

In total, 51 countries participating in neuromuscular blocker research were evaluated, and 29 countries met the criterion of having at least five collaborative links (Figure 4). The resulting network visualization delineated several clusters, with the largest connected component comprising 27 countries and additional clusters containing groups of at least 2 countries. An analysis of the network reveals that the USA functions as a central hub, establishing robust collaborations with Germany (represented in the purple cluster in Europe) and France (in the red cluster). Furthermore, the USA is actively engaged in scientific collaborations with South Korea and the People’s Republic of China, which are grouped within the yellow cluster representing Asian nations.

### 3.3. Journal Analysis

A “Journal Productivity and Citation Impact” analysis was performed to determine the most productive and frequently cited journals in the field of neuromuscular blockers and antagonists (Table 3). The analysis revealed that the *British Journal of Anaesthesia* was the most productive journal, accounting for 8.81% of the publications, and exhibited the highest impact, with a citation share of 18.52%. Other journals demonstrating significant productivity included the *European Journal of Anaesthesiology* (6.01%), the *Journal of Clinical Anesthesia* (5.01%), and *Acta Anaesthesiologica Scandinavica* (4.60%). Regarding citation counts, the *British Journal of Anaesthesia* was positioned as the journal with the greatest impact, while *Anesthesia & Analgesia* (8.10%) and the *European Journal of Anaesthesiology* (7.30%) ranked among the most frequently cited.

Co-citation relationships among journals were analyzed using VOSviewer, and 29 journals that received at least 50 citations were included in the evaluation (Figure 5). The results indicated that *Anesthesiology*, the *British Journal of Anaesthesia*, and the *European Journal of Anaesthesiology* occupied central positions within the network. Journals such as *The Lancet*, *The New England Journal of Medicine*, *Allergy*, and the *Journal of Investigational Allergology and Clinical Immunology* were in the blue cluster and established strong connections with those in the green cluster. In contrast, journals such as *Anesthesia & Analgesia*, *Pediatric Anesthesia*, and *BMC Anesthesiology* were positioned in the red cluster and demonstrated close collaboration with the green cluster.

### 3.4. Analysis of Authors

The most productive and most frequently cited authors in neuromuscular blocker and antagonist research were identified (Table 4). According to the analysis, Mertes P.M. emerged as the most productive author, with 10 publications (2.00%), followed by Fuchs-Buder T. with 9 publications (1.80%) and Blobner M. with 6 publications (1.20%). In terms of citation counts, Mertes P.M. demonstrated the highest impact, with 1273 citations (2.37%), while Demoly P. (553 citations, 1.02%) and Malinovsky J.M. (476 citations, 0.88%) ranked second and third, respectively.

To analyze scientific collaborations among authors, a “Co-Authorship Network in NMBAs Research” analysis was performed using VOSviewer (Figure 6). Based on Web of Science data, 196 authors with at least two collaborative publications were evaluated, and the collaboration network among these authors was visualized. The analysis revealed that 24 authors formed a central collaboration network. At the core of the collaboration network, Vera Saldien and Martine E. Prins, positioned in the red cluster, played a critical bridging role in NMBAs research. Lars I. Eriksson and Andreas Ekman, located in the green cluster, established strong collaborations with other researchers, significantly contributing to the field. Additionally, Bertrand Debaene and Claude Meistelman, through robust connections with different research groups in the blue and purple clusters, further contributed to the expansion of the scientific collaboration network.

### 3.5. Analysis of Publications

The publications with the highest citation counts were systematically evaluated and ranked (Table 5). The analysis revealed that the publication by Mertes P.M. and colleagues, published in 2011 in the *Journal of Allergy and Clinical Immunology* and addressing guidelines for anaphylaxis management during anesthesia, achieved the highest impact, with 282 citations. In second place, another publication by Mertes P.M., published in 2005 and focused on clinical guidelines for the management of anaphylaxis, garnered 273 citations, positioning it among the most influential studies. In third place, the study by Laxenaire M.C., published in 2001 and evaluating the frequency of anaphylactic reactions, distinguished itself with 253 citations. The study by Blobner M., published in 2010, examining the role of sugammadex in neuromuscular blockade reversal, recorded 174 citations, marking it as a significant contribution to the field. Additionally, the publication by Klein A.A., released in 2021, addressing guidelines for general anesthesia and recovery, attracted 170 citations, underscoring its relevance in the domain. Earlier high-impact studies, such as those published between 2001 and 2011, primarily focused on the management of anaphylaxis during anesthesia and the development of clinical practice guidelines (e.g., works by Mertes P.M. and Laxenaire M.C.). In contrast, more recent influential publications (2015–2021) have emphasized topics such as the reversal of neuromuscular blockade using agents like sugammadex, and updated guidelines for intraoperative monitoring and recovery practices (e.g., Klein A.A., 2021; Schuller P.J., 2015).

### 3.6. Co-Citation Analysis

The co-citation analysis performed using CiteSpace aimed to identify the prominent themes and scientific focal points in research on neuromuscular blockers and antagonists (Figure 7). The blue cluster focuses on topics such as anaphylaxis management during anesthesia, the measurement of tryptase levels, and the use of the bispectral index (BIS) for monitoring anesthesia depth. The yellow and orange clusters encompass studies related to the safety and efficacy of agents such as sevoflurane and alfentanil. The green cluster underscores the critical role of neuromuscular blockade safety and monitoring in surgical processes, whereas the red cluster highlights the importance of neuromuscular monitoring in the overall management of general anesthesia and surgical procedures.

### 3.7. Analysis of Citation Burst

The “Top 20 References with the Strongest Citation Bursts” analysis performed using CiteSpace identified pivotal studies that significantly influenced neuromuscular blocker and antagonist research across various periods (Figure 8). In the early 2000s (2000–2003), the study by Viby-Mogensen J. (1996) exhibited the most significant citation burst. Additionally, the works by Andrews JI (1999), Lowry D.W. (1998), and Vivien B. (2003) maintained high citation frequencies, reflecting their sustained impact on the field. Between 2007 and 2012, the studies by Gijsenbergh F. (2005) and Fuchs-Buder T. (2007) attracted considerable attention and established themselves as key references. More recently, during the 2020–2024 period, studies by Hristovska A.M. (2018, 2017) and Kheterpal S. (2020) emerged as the most highly cited, underscoring their significant influence on research related to neuromuscular blockers.

### 3.8. Analysis of Keywords Co-Occurrence Network

Keywords related to neuromuscular blockers and antagonists were analyzed to determine trends in the scientific literature (Figure 9). The terms were standardized, with singular and plural differences resolved, and only keywords that co-occurred at least five times were included in the analysis. Based on the results, the keyword clusters were categorized into three main themes. The red cluster represents general anesthesia, induction agents, and intubation processes, with key terms including “Anesthesia”, “Rocuronium”, “Propofol”, and “Tracheal Intubation”. The green cluster focuses on the pharmacological properties of neuromuscular blockers, highlighting keywords such as “Vecuronium”, “Atracurium”, “Pharmacokinetics”, and “Pharmacodynamics”. The blue cluster encompasses research on the reversal of neuromuscular blockade and the efficacy of antagonists, with “Reversal”, “Sugammadex”, and “Neostigmine” identified as the most frequently used keywords.

### 3.9. Analysis of Sankey Diagrams

Sankey diagrams were utilized to visualize the continuity and evolution of keywords over time (Figure 10). The analysis indicated that fundamental research topics such as “Anesthesia” and “Neuromuscular Blockade” have consistently remained prominent since 2000. During the period from 2000 to 2007, terms such as “Rocuronium” and “Pharmacokinetics” were predominant. From 2008 to 2015, topics such as “Safety” and “Monitoring” gained prominence in research. After 2016, “Sugammadex” and “Efficacy” emerged as key areas of investigation in modern neuromuscular blockade management.

### 3.10. Analysis of Keyword Citation Burst

The “Top 25 Keywords with Strongest Citation Bursts” analysis performed using CiteSpace revealed that certain keywords experienced significant citation bursts over time (Figure 11). In the Early Period (2000–2010), research was predominantly focused on the pharmacological properties of neuromuscular blockers, with keywords such as “Vecuronium”, “Atracurium”, “Halothane”, and “Succinylcholine” exhibiting the most pronounced citation bursts. During the Middle Period (2011–2020), “Sugammadex”, “Sevoflurane”, and “General Anesthesia” were frequently cited, largely due to the revolutionary impact of sugammadex. In the Recent Period (2021–2024), keywords such as “Neostigmine”, “Intubation”, “Surgery”, and “Efficacy” have increasingly garnered attention, reflecting their growing use in modern anesthetic practices.

### 3.11. Clustering and Centrality Metrics in Analysis of Keywords

Co-word network analysis reveals three main thematic clusters in the literature on neuromuscular blockers and their clinical use. Cluster 1 consists of immunological responses and safety-related terms. In this cluster, the most prominent terms are “neuromuscular blocking agents” (betweenness: 84.804; closeness: 0.015; PageRank: 0.056) and “anaphylaxis” (betweenness: 9.585; closeness: 0.013; PageRank: 0.042), which highlight perioperative issues. Additionally, terms such as “tryptase”, “skin test”, “latex”, and “antibiotics” are also included in this cluster and point to the diagnostic evaluation of these occurrences.

Cluster 2 focuses on anesthetic drugs and pharmacological agents. The most central concept in this cluster is “anesthesia” (betweenness: 484.510; closeness: 0.019; PageRank: 0.158). Other key terms such as “neuromuscular block” (betweenness: 144.938) and “anesthetics” (betweenness: 166.740) hold a central position in this field. “Sugammadex” (PageRank: 0.048; closeness: 0.013) is associated with pharmacological reversal strategies, while terms like “propofol”, “neostigmine”, “cisatracurium”, “remifentanil”, “general anesthesia”, “monitoring”, and “bispectral index” reflect the anesthetic agents and monitoring outcomes used in anesthesia practice.

Cluster 3 focuses on “succinylcholine”, a traditional agent known for its rapid onset of action. This term has a betweenness of 4.475, closeness of 0.012, and a PageRank of 0.015, and is specifically addressed within the scope of rapid sequence intubation protocols. When this analysis is considered overall, it is seen that Cluster 1 focuses on concerns related to immunology and safety, Cluster 2 on pharmacological agents and anesthesia management, and Cluster 3 on traditional agents with rapid onset of action (Table 6).

## 4. Discussion

This study comprehensively evaluated scientific research on NMBAs and their antagonists conducted between 2000 and 2024 using bibliometric analysis, thereby assessing global productivity, citation impact, and scientific collaboration networks. The 499 analyzed articles, published by 2424 authors from 51 countries and 850 different institutions, reveal the distribution and trends in scientific output within this field. To the best of our knowledge, this represents the first bibliometric analysis examining research patterns and collaborations in this specialized domain of anesthesiology.

### 4.1. Current Status and Global Collaborations

The findings indicate significant disparities in both productivity and impact among countries conducting research on neuromuscular blockers and antagonists. The USA stands out as the central country in terms of both publication output and international collaboration. Network analyses demonstrate that the USA has established strong partnerships with European countries (e.g., France and Germany) and Asian countries (e.g., China and South Korea), positioning it as a central hub in the global scientific communication network. These findings are consistent with the bibliometric analyses performed by Alfouzan et al. (2023) in the field of anesthesiology [21].

France, accounting for 19.04% of total citations, emerges as the most influential country in neuromuscular blocker research. This underscores the high scientific quality of French studies and their substantial citation impact. As shown in the Country Collaboration Network analysis, while France demonstrates strong research productivity, its collaborative connections are predominantly with other European nations, indicating potential opportunities for expanding research partnerships beyond regional boundaries.

Both China and South Korea have demonstrated increasing scientific productivity in neuromuscular blocker research. Our analysis indicates that China’s scientific output in the field of anesthesiology has significantly risen in recent years, a finding that is consistent with other studies highlighting increased contributions from China in related subfields [22,23]. Turkey has also contributed significantly to the scientific advancement in this domain, ranking highly in both productivity and citation metrics. It is anticipated that Turkey’s scientific collaborations, particularly with European countries, may evolve into broader global partnerships in the future. Variation in publication activity across countries may be due to a variety of systemic and structural barriers, including limited institutional research infrastructure (e.g., insufficient time, funding, or facilities), a lack of academic leadership in anesthesiology (e.g., inexperienced or clinically focused staff), a lack of structured incentives for research productivity (e.g., recognition, promotion pathways), and recruitment practices that prioritize clinical service over academic engagement.

### 4.2. Journal Analysis and Scientific Impact

The dominance of journals such as the *British Journal of Anaesthesia*, *Anesthesiology*, and *Anesthesia & Analgesia* in our study aligns with findings from previous bibliometric analyses in various subfields of anesthesiology. For instance, *Anesthesia & Analgesia* and the *British Journal of Anaesthesia* were among the top three most productive journals in studies focusing on postoperative pain management following cesarean section [24]. Similarly, in bibliometric research on ropivacaine use in postoperative pain, *Anesthesia & Analgesia* again ranked first in publication count, with the *British Journal of Anaesthesia* also ranking among the top three; moreover, the *Journal of Anesthesiology* was identified as having the highest average number of citations per article [25]. These journals are ranked among those with the highest impact factors in anesthesiology and pharmacology [26]. Additionally, in a bibliometric analysis of pediatric anesthesia, *Anesthesiology* was home to the most-cited article in the field, published by Gross et al. in 2002 [27]. These findings demonstrate that the influence of these journals extends across multiple domains of anesthesiology, reinforcing their prominent role as leading platforms for high-impact scientific contributions—both in our study and across the broader literature. The robust connections with general medical journals such as *The Lancet* and *The New England Journal of Medicine* suggest that research on neuromuscular blockers extends beyond the confines of anesthesiology and is evaluated within a comprehensive multidisciplinary framework. In addition, multidisciplinary journals with high citation impacts, such as the *Journal of Investigational Allergology and Clinical Immunology* and *Allergy*, further expand the scope of anesthesiology research by reinforcing interdisciplinary scientific connections. These results highlight the critical importance of interdisciplinary collaboration in neuromuscular blocker research and demonstrate that studies in anesthesiology are increasingly integrated with other medical disciplines.

### 4.3. Analysis of Authors’ Productivity and Scientific Collaborations

Examination of the most productive authors in neuromuscular blocker and antagonist research is crucial for understanding scientific progress in this field. Mertes PM emerges as one of the most influential researchers in perioperative anaphylaxis and the association between neuromuscular blockers and allergic reactions. His work has demonstrated that NMBAs are a primary cause of perioperative anaphylaxis, with these reactions typically being triggered by IgE-mediated mast cell and basophil activation. In particular, it has been shown that quaternary ammonium (QA) groups constitute the main allergenic structure in NMBAs, and that the use of pholcodine may increase sensitivity to these agents [28]. Moreover, Mertes PM’s studies have revealed that rocuronium and succinylcholine pose a higher risk of anaphylaxis compared to other NMBAs [29,30]. Some research on sugammadex has also underscored the potential for allergic reactions associated with this agent, noting an increase in NMBAs-related adverse reactions between 2000 and 2012. Mertes PM’s scientific impact extends beyond his highly cited publications; he has also facilitated interdisciplinary knowledge exchange through extensive international collaborations.

Vera Saldien is also recognized among the most influential authors due to her comprehensive studies on neuromuscular blockers and antagonists, positioning her at the center of scientific collaborations. Her research on the efficacy and safety profile of sugammadex in reversing neuromuscular blockade is particularly noteworthy [31]. In addition, a study evaluating the effects of various reversal strategies on respiratory muscle activity demonstrated that neostigmine adversely affects diaphragm and intercostal muscle function, potentially leading to respiratory complications [32].

Sugammadex has been shown to rapidly reverse deep blockade induced by rocuronium and vecuronium, although low doses may carry a risk of the reoccurrence of blockade [33]. Studies investigating the pharmacokinetic properties of sugammadex have indicated that this agent is rapidly eliminated via renal clearance and urinary excretion, and that it exhibits similar efficacy under both propofol and sevoflurane anesthesia—with a slightly superior safety profile observed with propofol [34].

Collectively, these findings highlight the pivotal role of authors like Vera Saldien and Mertes PM in advancing the field of neuromuscular blockade reversal and perioperative management, underscoring the importance of robust scientific collaborations in driving interdisciplinary research.

### 4.4. Scientific Impact of Most Highly Cited Publications

The most highly cited publications in neuromuscular blocker and anesthesia research are critical for revealing the theoretical and clinical interests and perspectives of researchers in this field. Notably, a study conducted by Mertes PM and colleagues, published in the *Journal of Allergy and Clinical Immunology*, examined eight years of anaphylaxis cases during anesthesia in France. This work demonstrated that such reactions occur more frequently than previously anticipated, with an elevated risk observed particularly in female patients [35]. Another study by Mertes PM revealed that a large proportion of perioperative allergic reactions are IgE-mediated, and that neuromuscular blockers, along with agents such as latex, hypnotics, and antibiotics, are among the most common triggers of anaphylaxis [36]. The study also emphasized the need for both immediate and secondary investigations, highlighting the diagnostic importance of skin tests and laboratory assessments (including tryptase, histamine, and specific IgE measurements). Similarly, a study by Laxenaire MC and Mertes PM, published in the *British Journal of Anaesthesia*, identified neuromuscular blocking agents as one of the leading causes of perioperative anaphylaxis [37]. In contrast, a randomized controlled trial by Blobner M and colleagues, published in the *European Journal of Anaesthesiology*, demonstrated that sugammadex reverses neuromuscular blockade significantly faster and more effectively than neostigmine [38].

The importance of research aimed at improving the safe use of neuromuscular blockers in anesthetic practice is further supported by the guidelines developed by Klein AA and colleagues, which advocate for the standardization of neuromuscular monitoring during anesthesia [39]. These highly cited studies indicate that research on neuromuscular blockers and antagonists has a direct impact on clinical management, with scientific collaborations driving advances in the field. The analysis of these publications reveals a concentrated focus on critical clinical topics such as the safe use of neuromuscular blockers and the management of anaphylaxis, all centered around the goal of enhancing patient safety. These findings underscore the need for future studies to prioritize safety- and innovation-oriented approaches, particularly through further research into personalized NMBA dosing, monitoring systems, and pharmacological safety.

### 4.5. Research Trends

Co-citation analysis is a valuable method for identifying the most influential and foundational studies in a field, delineating key themes, assessing scientific progress, and defining the boundaries of a research area [40]. The most prominent research topics identified include “neuromuscular block”, “neuromuscular monitoring”, “safety”, and “anaphylaxis”. This indicates that investigations focusing on the safety profiles of neuromuscular blockers, the requirements for effective monitoring, and the potential for adverse reactions have emerged as key areas of interest.

The longitudinal analysis of research topics visually illustrates the evolution of these subjects and highlights how critical aspects of clinical practice have transformed over time [17]. Citation burst analyses in research on neuromuscular blockers and antagonists have revealed significant shifts and evolving focal points within the field. Early research primarily concentrated on the pharmacodynamic properties and dosing strategies of neuromuscular blockers; in particular, studies by Viby-Mogensen (1996) and Andrews (1999) provided essential foundational insights into the safe use of these agents [41,42,43]. However, the predominantly pharmacological focus of these early studies resulted in certain limitations in their clinical applicability. In the post-2010 era, research has increasingly shifted towards clinical efficacy, patient safety, and optimization [44,45]. This transition has paralleled the integration of advanced monitoring systems into clinical practice and the development of novel reversal agents. For example, studies by Gijsenbergh (2005) and Fuchs-Buder (2007) underscore the growing clinical need for effective and safe reversal of neuromuscular blockade [45,46,47].

Personalized dosing strategies and artificial intelligence-supported monitoring systems for the management of neuromuscular blockade have recently come to the forefront [48,49]. Studies by Hristovska (2018) and Kheterpal (2020) have demonstrated that individualized anesthesia management has the potential to improve clinical outcomes [48,49,50]. However, these approaches require further validation through large-scale, multicenter, randomized controlled trials. The integration of AI-based dose optimization and advanced neuromuscular monitoring systems could significantly reduce the risk of postoperative residual neuromuscular blockade. Nonetheless, the cost-effectiveness, feasibility, and long-term impact of these technologies on patient outcomes remain subjects of ongoing debate [51,52]. Future research should focus on evaluating the clinical differences between standard neuromuscular blockade management protocols and personalized approaches.

### 4.6. Thematic Shifts and Clinical Implications

This study comprehensively elucidates the thematic evolution in research on neuromuscular blockers and their antagonists. Keyword analysis reveals that scientific priorities in this field have shifted over time, becoming directly linked to clinical applications. In particular, the growing interest in general anesthesia and induction agents supports the central role of neuromuscular blockers in airway management and surgical anesthesia.

Our Sankey diagram analysis further reveals that the scientific priorities within neuromuscular blocker research have evolved over time. While early studies focused on the pharmacological profiles of neuromuscular blocking agents, more recent research has shifted towards strategies that prioritize patient safety. The increased use of agents such as sugammadex and neostigmine reflects a growing need for the effective reversal of neuromuscular blockade. Future studies should focus on optimizing personalized neuromuscular blockade management, advanced monitoring techniques, and pharmacological reversal approaches to provide safer and more effective solutions in anesthetic practice.

This study has several limitations that should be noted. The WoSCC was selected as the sole data source due to its high-quality and standardized indexing, which ensured the reliability and consistency of the bibliometric data. However, it should be acknowledged that this choice may have excluded relevant articles indexed in other databases such as Scopus or PubMed, potentially leading to a partial representation of the global research landscape on neuromuscular blockers and their antagonists. Additionally, the analysis is based on citation counts and co-authorship networks, which may not fully reflect the clinical impact or translational value of individual studies, with citation biases and self-citations possibly affecting the interpretation of scientific contributions. Language limitations also exist, as only articles published in English were included, thereby potentially overlooking valuable findings published in other languages. Moreover, citation-based indicators may reflect biases related to variations in journal impact factors, publication age, and database coverage, all of which may influence the perceived importance of specific studies or authors. Finally, although the study covers a twenty-year period (2000–2024), emerging trends in recent years may not be fully captured in the data.

## 5. Conclusions

This bibliometric analysis examined global research on NMBAs and their antagonists in the field of anesthesia from 2000 to 2024, analyzing co-citation networks, key contributing researchers, and thematic developments. The findings indicate a marked increase in studies on neuromuscular monitoring and reversal agents, such as sugammadex, over the past two decades. The United States, France, and China emerged as the most contributory countries in NMBAs research, with their extensive international collaborations playing a pivotal role in shaping scientific progress. Highly influential studies have predominantly focused on NMBA pharmacokinetics, safety, anaphylaxis risks, and the clinical benefits of sugammadex, underscoring its critical role in reducing residual neuromuscular blockade (rNMB) and enhancing patient safety. Keyword trend analysis revealed a shift toward precision medicine approaches, including personalized NMB management, advanced monitoring technologies, and pharmacological optimization.

Future studies should focus on the development of personalized anesthesia protocols supported by neuromuscular monitoring technologies. Interdisciplinary collaboration and technological innovation will support progress in this field and play a key role in enhancing patient safety. Study findings indicate that research and international collaborations on neuromuscular blockers remain limited in low- and middle-income countries; in particular, policymakers in these countries should increase support for scientific research in the field of anesthesiology.

## Figures and Tables

**Figure 1 healthcare-13-01146-f001:**
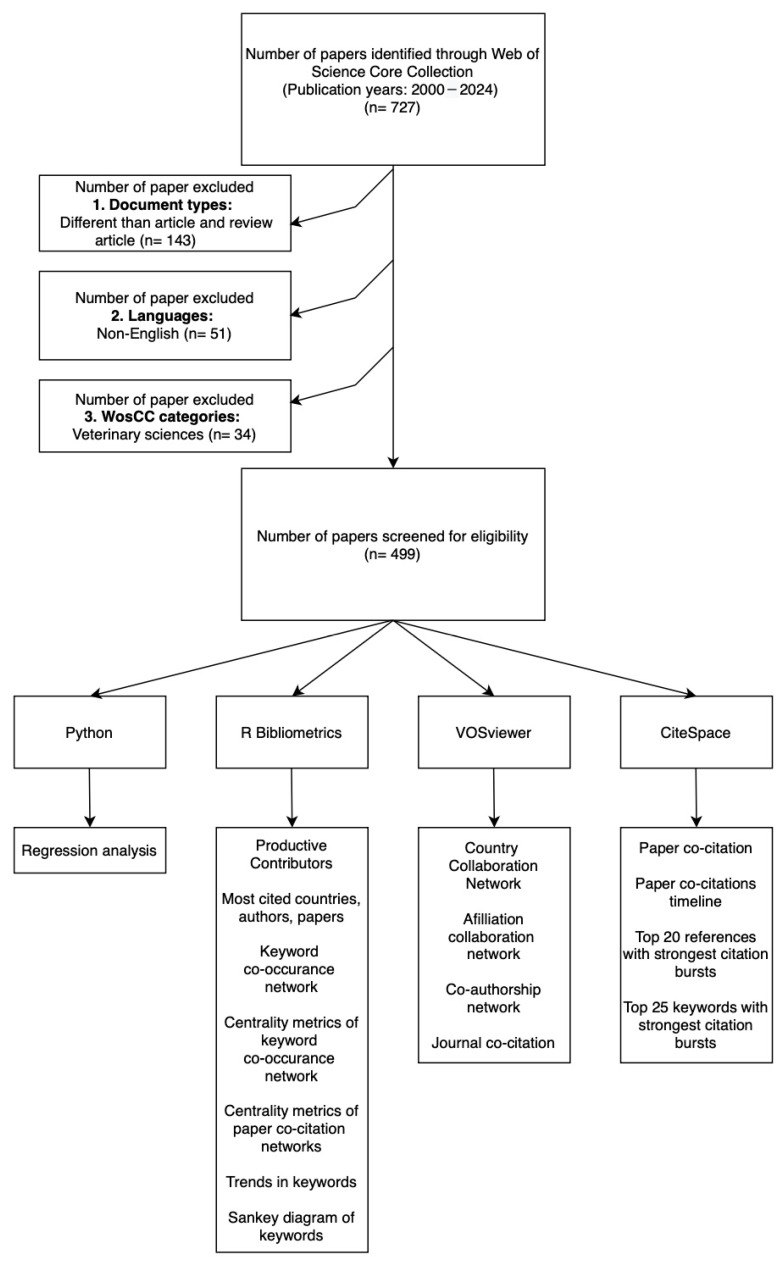
Flowchart.

**Figure 2 healthcare-13-01146-f002:**
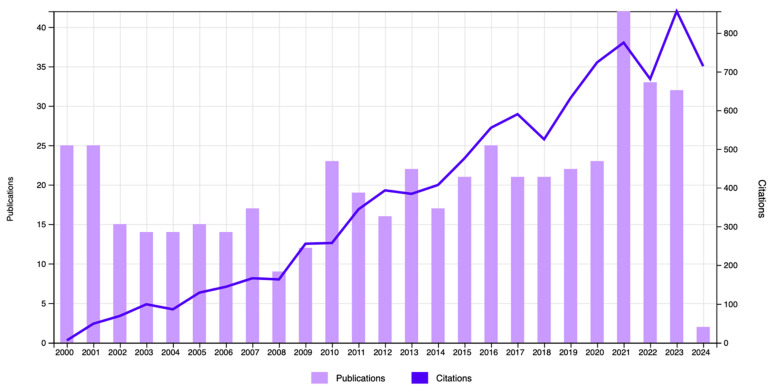
The number of times cited and the number of publications over time. The citations in this graph represent the number of citations received in each individual year. The publication bars indicate the number of articles published per year within the same dataset.

**Figure 3 healthcare-13-01146-f003:**
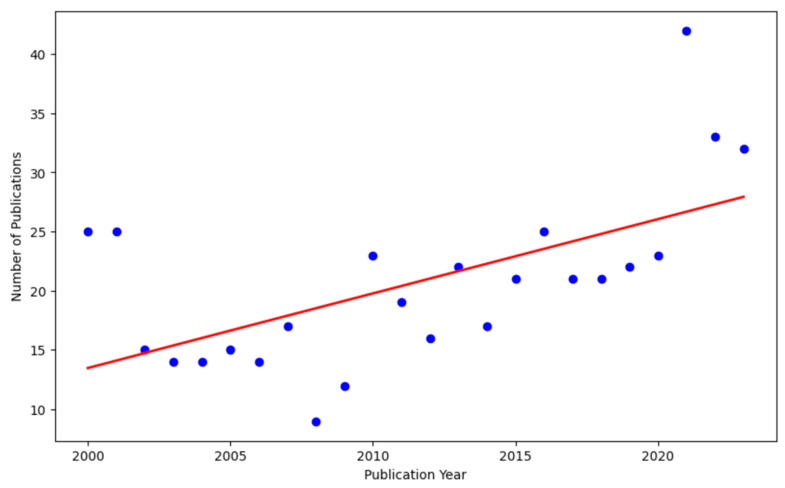
The number of publications by year, with a linear regression analysis. The plot was generated using the Python sklearn and matplotlib libraries. The data cover the publication years from 2000 to 2024, with the blue dots representing the annual publication counts and the red line depicting the linear regression trend. The position of each dot indicates the yearly total for publications, while the slope of the regression line illustrates the overall upward trajectory in publication frequency.

**Figure 4 healthcare-13-01146-f004:**
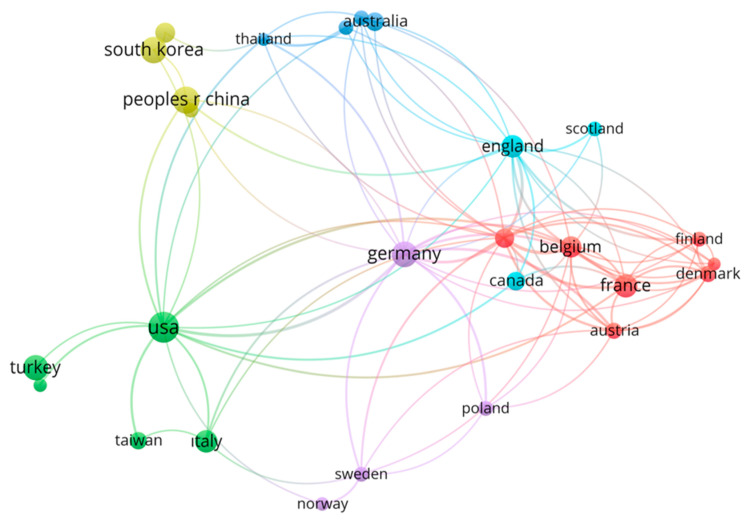
Country Collaboration Network in NMBAs research. Visualization created using VOSviewer software. Nodes represent individual countries involved in research collaboration, with node size corresponding to frequency of country’s research output. Connecting lines illustrate collaborative relationships between countries, with thicker links indicating stronger cooperation. Colors indicate clusters of countries with stronger intra-group collaboration links, as identified by VOSviewer’s clustering algorithm.

**Figure 5 healthcare-13-01146-f005:**
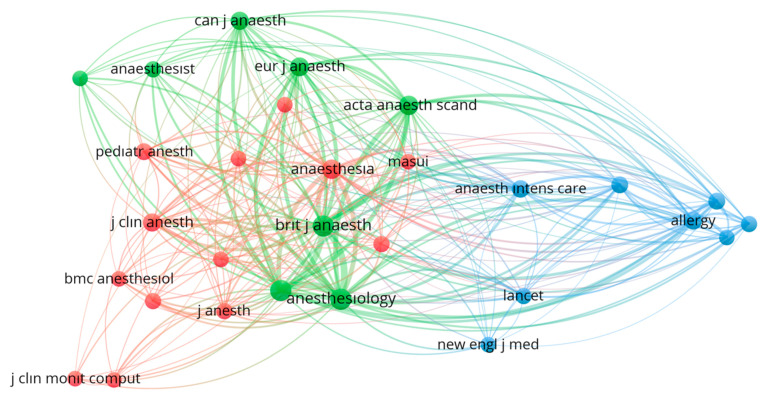
The Journal Co-Citation Network in NMBAs research. The visualization was generated with VOSviewer software. Each node represents an academic journal, with the size indicating the frequency of published articles from the dataset. The links between nodes illustrate citation or co-citation relationships between journals; thicker lines suggest stronger connections. The journals are grouped into color-coded clusters based on citation patterns, indicating related research areas or publication themes within the field of anesthesiology and related specialties.

**Figure 6 healthcare-13-01146-f006:**
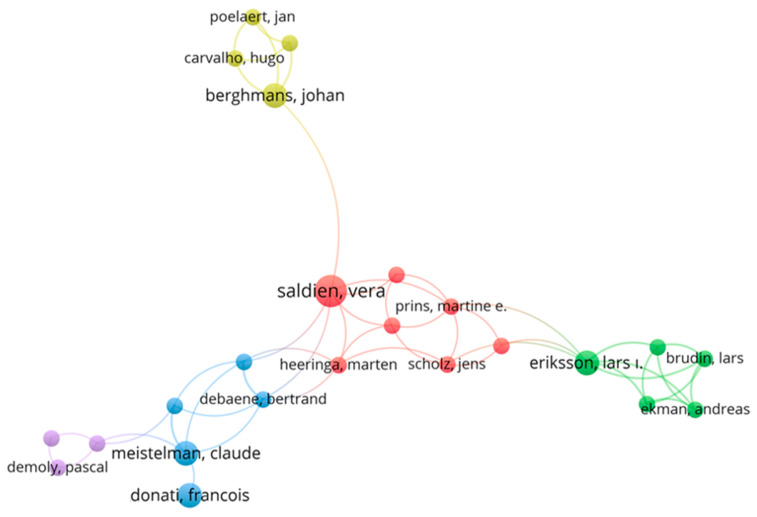
The Co-Authorship Network in NMBAs Research. The visual representation of co-authorship networks was generated by VOSviewer. Each node represents an individual author, while the links indicate co-authorship links. The colors group authors into clusters who collaborate more frequently. The thickness of the links reflects the strength of cooperation or shared publications, and the node size is proportional to each author’s prominence in the network.

**Figure 7 healthcare-13-01146-f007:**
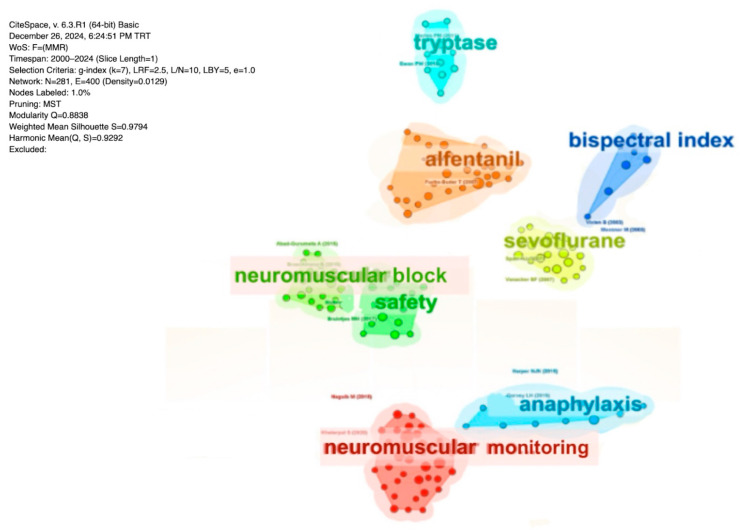
Paper co-citations. Visualization of paper clusters produced by CiteSpace. Each colored region represents cluster of highly co-cited papers; node size reflects citation frequency; links indicate co-citation relationships.

**Figure 8 healthcare-13-01146-f008:**
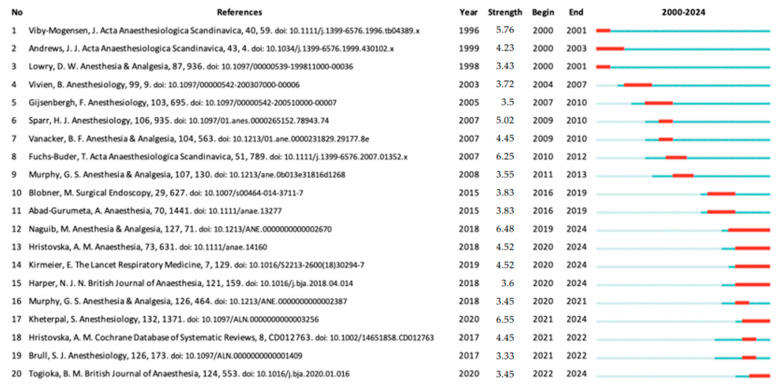
The Top 20 References with the Strongest Citation Bursts. A display of the top 20 references with the strongest citation bursts from 2000 to 2024, as identified by CiteSpace. Each row lists the reference, its burst strength, and the time interval (beginning and end years). The colored bars illustrate the duration of the burst period (in red) against the broader timeline (in blue).

**Figure 9 healthcare-13-01146-f009:**
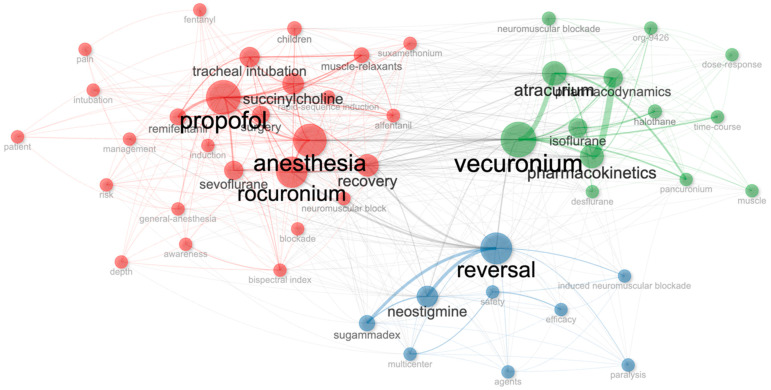
Keywords co-occurrence network. Network visualization of anesthesia-related keywords, with node size corresponding to frequency of keyword occurrence. Links represent co-occurrences of terms in the literature, and color-coded clusters highlight distinct thematic groupings. Map illustrates key research areas and connections among anesthetic agents and techniques.

**Figure 10 healthcare-13-01146-f010:**
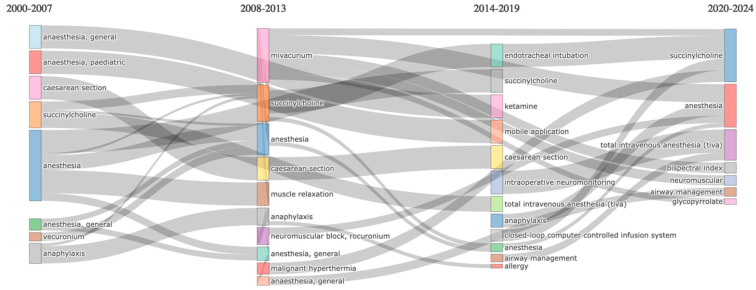
The Sankey diagram of keywords. A Sankey diagram tracing the evolution of anesthesia-related topics over four time periods. Each colored block in a given interval represents a key theme in the diagram, and the connecting flows show how research focus shifts or overlaps across subsequent periods. The thickness of each flow reflects the relative frequency or strength of the connection, illustrating how certain anesthesia subjects emerge, persist, or decline over time.

**Figure 11 healthcare-13-01146-f011:**
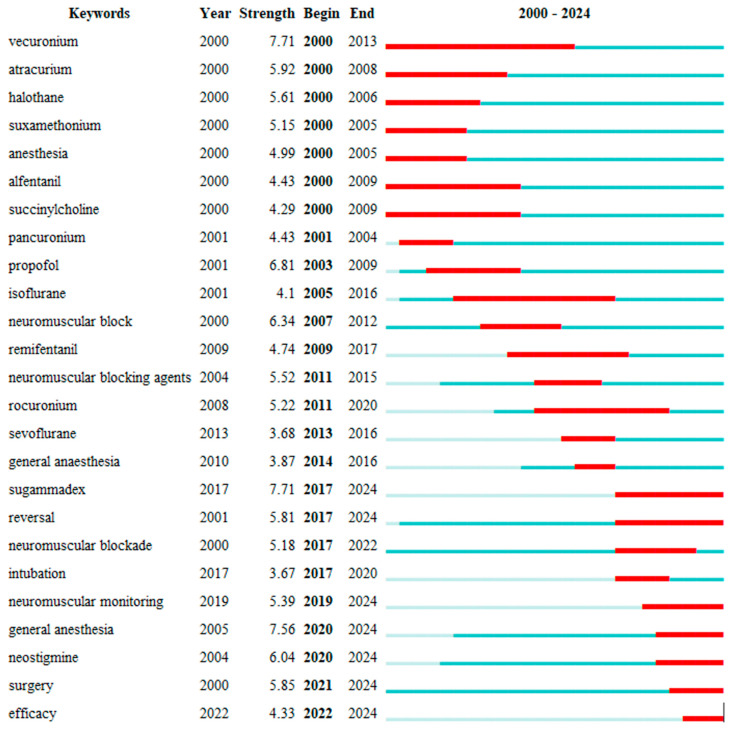
The top 25 keywords with the strongest citation bursts. A list of the top 25 anesthesia-related keywords exhibiting the strongest citation bursts between 2000 and 2024, as determined by CiteSpace. Each row shows the keyword, its burst strength, and the start/end years of the citation surge. The red portion of each timeline indicates the burst period, while the blue portion reflects the full-time range under study.

**Table 1 healthcare-13-01146-t001:** Search strategy for Web of Science database.

Component	Details
Database	WoSCC (SCI-Expanded)
Search Date	20 September 2024
Search Fields	Title (TI) and Author Keywords (AK)
Keywords Used 1	“Neuromuscular block” OR “Neuromuscular monitoring” OR “Neuromuscular management” OR “Depolarizing agents” OR “Non-depolarizing agents” OR “Cholinesterase inhibitors” OR “Neuromuscular antagonists” OR “Succinylcholine” OR “Pancuronium” OR “Vecuronium” OR “Rocuronium” OR “Atracurium” OR “Cisatracurium” OR “Mivacurium” OR “Sugammadex” OR “Neostigmine” OR “Pyridostigmine” OR “Edrophonium”
Keywords Used 2	“Anesthesia” OR “Anaesthesia” OR “Anesthesiology”
Boolean Combination	Keywords Used 1 AND Keywords Used 2
Time Frame	2000–2024
Language	English only
Document Types	Articles and review articles
Exclusion Criteria	Veterinary publications, duplicate records, incomplete bibliographic data, and non-English articles
Keyword Selection	Based on expert consensus among anesthesiologists with ≥10 years of experience

Web of Science Core Collection (WoSCC), Science Citation Index Expanded (SCI-Expanded).

**Table 2 healthcare-13-01146-t002:** Productivity and citations of countries in neuromuscular blocker and antagonist research.

	Top 10 Most Productive Contributing Countries		Top 10 Active Countries According to Number of Citations	
	Country	N (%)	Country	N(%)
1	USA	78 (15.63)	France	1776 (19.04)
2	People’s Republic of China	50 (10.02)	USA	1055 (11.31)
3	South Korea	46 (9.21)	Germany	908 (9.73)
4	Germany	44 (8.81)	United Kingdom	812 (8.70)
5	Turkey	39 (7.81)	Belgium	575 (6.16)
6	France	30 (6.01)	Korea	483 (5.18)
7	England	29 (5.81)	Italy	428 (4.59)
8	Italy	29 (5.81)	Denmark	398 (4.26)
9	Belgium	23 (4.60)	Turkey	308 (3.30)
10	Japan	21 (4.20)	Australia	306 (3.28)

United States of America (USA).

**Table 3 healthcare-13-01146-t003:** Journal productivity and citation impact in anesthesiology research.

	Top 10 Most Productive Contributor Journals		Top 10 Active Journals According to Number of Citations	
	Journal	N (%)	Journal	N(%)
1	*British Journal of Anaesthesia*	44 (8.81)	*British Journal of Anaesthesia*	1728 (18.52)
2	*European Journal of Anaesthesiology*	30 (6.01)	*Anesthesia & Analgesia*	756 (8.10)
3	*Journal of Clinical Anesthesia*	25 (5.01)	*European Journal of Anaesthesiology*	681 (7.30)
4	*Acta Anaesthesiologica Scandinavica*	23 (4.60)	*Acta Anaesthesiologica Scandinavica*	520 (5.57)
5	*Pediatric Anesthesia*	20 (4.00)	*Journal of Investigational Allergology and Clinical Immunology*	424 (4.54)
6	*Anesthesia & Analgesia*	19 (3.80)	*Journal of Clinical Anesthesia*	395 (4.23)
7	*BMC Anesthesiology*	14 (2.80)	*Journal of Allergy and Clinical Immunology*	282 (3.02)
8	*Medicine*	12 (2.42)	*Anaesthesia*	271 (2.90)
9	*Journal of Anesthesia*	12 (2.42)	*Allergy*	223 (2.39)
10	*Anaesthesia and Intensive Care*	11 (2.22)	*Pediatric Anesthesia*	218 (2.33)

**Table 4 healthcare-13-01146-t004:** Top authors’ productivity and citation analysis in neuromuscular blocker and antagonist research.

	Top 10 Most Productive Contributing Authors		Top 10 Active Authors According to Number of Citations	
	Author	N (%)	Author	N (%)
1	Mertes P.M.	10 (2.00)	Mertes P.M.	1273 (2.37)
2	Fuchs-Buder T.	9 (1.80)	Demoly P.	553 (1.02)
3	Blobner M.	6 (2.20)	Malinovsky J.M.	476 (0.88)
4	Demoly P.	6 (1.20)	Aberer W.	424 (0.78)
5	Eikermann M.	6 (1.20)	Laxenaire M.M.	405 (0.75)
6	Adamus M.	5 (1.00)	Blobner M.	336 (0.62)
7	Donati F.	5 (1.00)	Alla F.	282 (0.52)
8	Malinovsky J.M.	5 (1.00)	Auroy Y.	283 (0.52)
9	Stourac P.	5 (1.00)	Jougla E.	284 (0.52)
10	Carron M.	4 (0.80)	Tréchot P.	285 (0.52)

**Table 5 healthcare-13-01146-t005:** Top 10 publications by average citation number (journal/first author/publication year/number of citations).

	Title	Journal	Journal Category Quartile	Journal Impact Factor (Five-Year)	First Author	Publication Year	Number of Citations
1	Mertes, Paul Michel et al. “Anaphylaxis during anesthesia in France: an 8-year national survey”. *Journal of Allergy and Clinical Immunology* vol. 128,2 (2011): 366–373. doi:10.1016/j.jaci.2011.03.003	*Journal of Allergy and Clinical Immunology*	Q1	10.2	Mertes, P.M.	2011	282
2	Mertes, P M et al. “Reducing the risk of anaphylaxis during anaesthesia: guidelines for clinical practice”. *Journal of Investigational Allergology and Clinical Immunology* vol. 15,2 (2005): 91–101.	*Journal of Investigational Allergology and Clinical Immunology*	Q1	6.5	Mertes, P.M.	2011	273
3	Laxenaire, M C et al. “Anaphylaxis during anaesthesia. Results of a two-year survey in France”. *British Journal of Anaesthesia* vol. 87,4 (2001): 549–558. doi:10.1093/bja/87.4.549	*British Journal of Anaesthesia*	Q1	9.5	Laxenaire, M.C.	2001	253
4	Blobner, Manfred et al. “Reversal of rocuronium-induced neuromuscular blockade with sugammadex compared with neostigmine during sevoflurane anaesthesia: results of a randomised, controlled trial”. *European Journal of Anaesthesiology* vol. 27,10 (2010): 874–881. doi:10.1097/EJA.0b013e32833d56b7	*European Journal of Anaesthesiology*	Q1	4.3	Blobner, M.	2010	174
5	Klein, A A et al. “Recommendations for standards of monitoring during anaesthesia and recovery 2021: Guideline from the Association of Anaesthetists”. *Anaesthesia* vol. 76,9 (2021): 1212–1223. doi:10.1111/anae.15501	Anaesthesia	Q1	8	Klein, A.A.	2021	170
6	Ebo, D G et al. “Anaphylaxis during anaesthesia: diagnostic approach”. *Allergy* vol. 62,5 (2007): 471–487. doi:10.1111/j.1398-9995.2007.01347.x	*Allergy*	Q1	11.8	Ebo, D.G.	2007	158
7	Mertes, P M et al. “Reducing the risk of anaphylaxis during anaesthesia: guidelines for clinical practice”. *Journal of Investigational Allergology and Clinical Immunology* vol. 15,2 (2005): 91–101.	*Journal of Investigational Allergology and Clinical Immunology*	Q1	6.5	Mertes, P.M.	2005	152
8	Schuller, P J et al. “Response of bispectral index to neuromuscular block in awake volunteers”. *British Journal of Anaesthesia* vol. 115 Suppl 1 (2015): i95–i103. doi:10.1093/bja/aev072	*British Journal of Anaesthesia*	Q1	9.5	Schuller, P.J.	2015	135
9	Ryu, J-H et al. “Effects of magnesium sulphate on intraoperative anaesthetic requirements and postoperative analgesia in gynaecology patients receiving total intravenous anaesthesia”. *British Journal of Anaesthesia* vol. 100,3 (2008): 397–403. doi:10.1093/bja/aem407	*British Journal of Anaesthesia*	Q1	9.5	Ryu, J.H.	2008	135
10	Ewan, P W et al. “BSACI guidelines for the investigation of suspected anaphylaxis during general anaesthesia”. *Clinical and Experimental Allergy: Journal of the British Society for Allergy and Clinical Immunology* vol. 40,1 (2010): 15–31. doi:10.1111/j.1365-2222.2009.03404.x	Clinical and Experimental Allergy	Q1	5.2	Ewan, P.W.	2010	134

Note: The Journal Quartile and Journal Impact Factor (JIF) values are based on the 2023 edition of the Journal Citation Reports (Clarivate Analytics).

**Table 6 healthcare-13-01146-t006:** Clustering and centrality metrics in analysis of keywords.

	Keyword	Cluster	Betweenness	Closeness	Page Rank
1	anesthesia	2	484.51	0.019	0.158
2	neuromuscular block	2	144.938	0.017	0.095
3	anesthetics	2	166.74	0.017	0.094
4	neuromuscular blocking agents	1	84.804	0.015	0.056
5	sugammadex	2	7.89	0.013	0.048
6	anaphylaxis	1	9.585	0.013	0.042
7	propofol	2	5.232	0.013	0.028
8	neostigmine	2	0.668	0.011	0.024
9	cisatracurium	2	2.753	0.013	0.022
10	tryptase	1	0.972	0.012	0.022
11	monitoring	2	3.171	0.012	0.021
12	remifentanil	2	1.705	0.012	0.018
13	skin test	1	0.278	0.012	0.018
14	latex	1	0.14	0.011	0.017
15	antibiotics	1	0.471	0.012	0.017
16	general anesthesia	2	4.251	0.012	0.017
17	bispectral index	2	2.507	0.012	0.016
18	succinylcholine	3	4.475	0.012	0.015
19	intubation	2	0.56	0.012	0.015
20	vecuronium	2	1.115	0.012	0.013

## Data Availability

The original data presented in the study were obtained from Web of Science.

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
