# Peer review of "Bibliometric Analysis of Research Trends and Global Collaborations in Anesthesia on Neuromuscular Blockers and Antagonists (2000–2024)"

_healthcare, 2025, doi:10.3390/healthcare13101146_

Round 1

Reviewer 1 Report

Comments and Suggestions for Authors

This manuscript presents a large-scale bibliometric analysis of global research on neuromuscular blockers and their antagonists over the past two decades. It aims to map international collaboration patterns, thematic trends, and influential contributors within the field.

The topic is important given the growing focus on patient safety, neuromuscular monitoring, and precision anesthesia. The authors utilize multiple bibliometric tools (R Bibliometrix, VOSviewer, CiteSpace) to provide multilayered insights. The study clearly defines its scope and goals.

While the topic is relevant and the dataset comprehensive, the manuscript requires substantial revisions before it can be considered for publication.

Major Concerns

The manuscript is overly long, with extensive repetitions, particularly in the Results and Discussion sections.

The search strategy is described vaguely. The full Boolean search query and rationale for inclusion/exclusion (e.g., why only English-language articles) should be provided.

Only one database (WoSCC) was used. The limitations of this decision should be discussed more explicitly.

The analysis often reads as a catalog of data points rather than a critical synthesis of trends. For example, country and author rankings are reported without contextualizing possible reasons (e.g., funding structures, historical leadership in anesthesiology, policy shifts). The Discussion would benefit from more interpretation of why trends exist and how they may influence future research.

The conclusion reiterates findings but does not propose specific, actionable recommendations for researchers or policymakers. 

Reviewer 2 Report

Comments and Suggestions for Authors

Introduction

  • Please discuss whether any previous bibliometric analyses have been conducted on this topic to highlight the novelty of this study.
  • Additionally, explain how bibliometric analysis contributes to enhancing clinical impact in the field of anesthesiology.

Methods

  • Presenting the Boolean search strategy in a structured table format would improve clarity and readability.
  • Provide a rationale for selecting only WoSCC as the data source, and discuss why Scopus or PubMed were not included, beyond mentioning this as a limitation in the Discussion section.

Results

  • Figure 2A: Clarify whether this represents the accumulation of citations.
  • Figure 2B: Report whether this model is statistically significant.
  • Table 2: Use a more formal approach when referring to country names.
  • Figures 3, 4, 5, and 6: Explain the meaning of different colors and clusters.
  • Table 6: Specify the year in which the journal quartile and Journal Impact Factor (JIF) were issued.
  • Discuss the significance of variations in research trends over time more explicitly.
  • Several figures are too small, making them difficult to interpret, especially for general readers. Consider enlarging them for better readability.

Discussion

  • Strengthen the discussion by comparing findings with previous bibliometric studies in anesthesiology to provide better context.
  • Include an explanation of potential biases in bibliometric studies, such as variations in journal impact factors, to improve transparency.

Reviewer 3 Report

Comments and Suggestions for Authors

This manuscript is a bibliometric analysis of research trends and collaboration networks on neuromuscular blockers (NMBs) and their antagonists worldwide from 2000 to 2024.

The primary method was through the Web of Science Core Collection.

This manuscript is not a research paper but more of a review.
It was not conducted using any experimental or new tools.
It is just a trend analysis.
I could not catch much of the novelty of this manuscript.

There are many typos in this manuscript. I believe that this manuscript is not yet refined and is not ready for publication.

The quality of the figures is very poor. The contents of the figures are not easily grasped and their meanings are not very specific.
In particular, the contents of Figure 8 are almost impossible to confirm.

The captions for each figure file are processed as graphics, not text, and are redundant.

I strongly believe that this manuscript should be changed to a review-type trend analysis.

Comments on the Quality of English Language

The English is very poor and there are many typos in the manuscript.

Therefore, the readability of the manuscript is very poor.

Round 2

Reviewer 1 Report

Comments and Suggestions for Authors

This manuscript presents a thorough and well-structured bibliometric analysis of global research trends, citation impact, and collaborative networks in the field of neuromuscular blocking agents (NMBAs) and their antagonists over a 24-year period. The authors have employed robust methodologies using established tools such as R bibliometrix, VOSviewer, and CiteSpace, and the study is supported by a comprehensive and meticulously described dataset extracted from the Web of Science Core Collection.

Author Response

Response to Reviewer 1–Round 2

Comment 1: This manuscript presents a thorough and well-structured bibliometric analysis of global research trends, citation impact, and collaborative networks in the field of neuromuscular blocking agents (NMBAs) and their antagonists over a 24-year period. The authors have employed robust methodologies using established tools such as R bibliometrix, VOSviewer, and CiteSpace, and the study is supported by a comprehensive and meticulously described dataset extracted from the Web of Science Core Collection.

Response 1: Thank you for your valuable and encouraging comment.